# Overall maternal morbidity during pregnancy using the WHO-WOICE tools

Hanane Hababa[1]*, Bouchra Assarag[2]

1 Regional Directorate of Health and Social Protection Marrakech-Safi Morocco, Marrakech, Morocco,
2 National School of Public Health Rabat Morocco, Rabat, Morocco

* hababahanane@gmail.com

## Abstract

### Introduction

The objective of this study is to evaluate the prevalence of nonsevere maternal morbidities (including overall health, domestic and sexual violence, functionality, and mental health) in women during antenatal care in rural versus urban areas. This study aimed to describe the factors that affect women's health during pregnancy by administration of the WHO's WOICE 2.0 instrument.

### Methods

This was a cross sectional study conducted at perinatal care health centers in Morocco (5 in rural and 5 in urban). We recruited n = 257 women in the third trimester of their pregnancy using a questionnaire developed by the WHO to assess maternal morbidity, which includes various instruments that measure different aspects of maternal health. This tool evaluates the functionality and ability to perform daily tasks includes a tool that evaluates mental health, the General Anxiety Disorder 7-item test (GAD-7), and the 9-item Patient Health Questionnaire (PHQ-9), to assess depression. Data on health conditions and socio-demographic characteristics were collected through structured interview, medical record review, and clinical examination. This paper presents descriptive data on the distribution of functioning status among antepartum women.

### Results

In all, 257 women at a mean age of 30 years were included, and the majority had a partner (98%) and primary education (68.48%). Over one third of the population cannot read. Exposure to violence occurred in 12.23%. Sexual dissatisfaction was reported by 28.20% of antepartum women. Overall, women reported very good and good health (39.69%), and more than third had a medical condition (39.69%). There was an overall rate of anxiety in 83.65%, depression in 43.57%. Based on the $\chi 2$ test, Fisher exact test, or Kruskal-Wallis test, there was a significant relationship between the distribution of depression (p<0.001) and sexual satisfaction (p<0.01) between urban and rural women.

**Data Availability Statement:** All relevant data are within the paper.

**Funding:** The authors received no specific funding for this work.

**Competing interests:** The authors have declared that no competing interests exist.

**Abbreviations:** APC, Antepartum care; AGAD7, General anxiety disorder 7; MNM, Maternal near miss; MMWG, Working group maternal morbidity; PHQ-9, Patient health questionnaire 9; SPSS, Statistical package for the social sciences; WHO, World health Organization; WHODAS, World health Organization disability assessment schedule 2.0.

## Conclusion

Considering these results, antepartum depression and anxiety were highly prevalent in our sample and contributed substantially to perceived disability. These serious threats to health must be further investigated and more data are needed to comprehensively quantify the problem in Morocco.

## Introduction

Antenatal care (ANC) is a key intervention to prevent women's morbidity and mortality. Improving maternal health and reducing maternal mortality have been key concerns of the international community for many years. The United Nations has set a goal to reduce maternal mortality by at least 70% by 2030 as part of its Sustainable Development Goals (SDGs) [1]. To achieve this goal, the international community has focused on improving access to quality maternal healthcare services, increasing investments in maternal health, and addressing the social and cultural factors that contribute to maternal mortality [2, 3]. In 2015, an estimated 303 000 women died during or after pregnancy [1, 4]. However, it's important to note that maternal mortality is only one aspect of the burden of ill-health related to inadequate maternal healthcare. Many women suffer from long-term health consequences due to inadequate care during pregnancy and childbirth [5]. The burden of maternal morbidity is not yet known [6]. WHO estimates that for every recorded maternal death, 20 to 30 women suffer morbidity. The World Health Organization (WHO) estimates that, in 2016 alone, maternal conditions contributed to around 19 million disability-adjusted life years lost [7]. Maternal morbidity can include physical health problems such as hemorrhage, infection, and high blood pressure, as well as mental health problems such as depression, and anxiety. These health problems can impact a woman's ability to function, including her cognition, mobility, and participation in society. They can also affect a woman's body image and socioeconomic status [7].

In the absence of standardized instruments for accurate assessment of overall and non-severe maternal morbidities, the WHO implemented the Maternal Morbidity Working Group (MMWG) in 2012 [2]. The latter defined maternal morbidity as "any health condition attributed to and/or aggravated by pregnancy and childbirth that has a negative impact on a woman's well-being" [8] and created an instrument, later called WOICE, for measuring maternal morbidity, focusing on health and a woman's self-perception of well-being [2, 6, 8]. The WHO-WOICE is a questionnaire developed to gather information directly from women about their experiences and perceptions of pregnancy and its potential long-term impacts. Up to now, the prevalence of nonsevere maternal morbidity among pregnant women remains largely unknown, especially the conditions related to domestic violence, sexual violence, and changes in mental health, social role, and functionality [7, 9, 10]. These issues may have a negative impact on women's lives. A lack of understanding of these issues may lead healthcare providers to dismiss their occurrence. As a result, this study was aimed at implementing the WHO working group (WOICE) tool in a middle-income setting in Morocco, to evaluate the whole prevalence of maternal morbidity among pregnancy, along with factors associated with clinical, mental, or functional impairment.

## Materials and methods

This cross-sectional study used a questionnaire developed by the WHO to assess maternal morbidity, which includes various instruments that measure different aspects of maternal

health. The maternal morbidity measurement questionnaire called WOICE includes several tools that have already been previously translated and adapted to French [8]. The review was conducted by experienced obstetric investigators and the version was tested to measure the time of application and then adapt and modify some words to ascertain accurate understanding. This several tools have already been adapted. Adapting these tools to the Moroccan context helps to ensure that the results are relevant and meaningful to the local population. WOICE includes the 12-item version of the World Health Organization's Disability Assessment Schedule (WHODAS 2.0), and also a tool that evaluates mental health, the General Anxiety Disorder 7-item test (GAD-7), and the 9-item Patient Health Questionnaire (PHQ-9), to assess depression, both already adapted to French [11, 12].

To measure substance use and abuse, WOICE includes Alcohol, Smoking and Substance Involvement Screening Test (ASSIST) [13, 14]. For sexual satisfaction and sexual and domestic violence, parts of some scores already validated are within the WOICE, such as the Brief Sexual Symptom Checklist for Women (BSSC-W) and some questions from a questionnaire used in the Multicountry Study on Women's Health and Domestic Violence against Women of the WHO [15].

A convenience sampling strategy used so that all eligible women invited to participate. Women were invited to participate if they are in their third trimester of pregnancy (28 or more weeks). The exclusion criteria are minors under 18 years of age and women suffering from a mental or physical disability. Data collection lasted two months at each site. Written informed consent was obtained, except in cases where the woman was illiterate in which case a witness confirmed the accurate reading of the consent form and a thumbprint of the participant was obtained. This procedure was approved by the ethics committees. This study was conducted in 10 health center in Marrakech Morocco (05 urban and 05 rural). The Inclusion criteria of this health centers are: Representation of the urban and rural areas and the number of perinatal visits exceeds 30 visits per month. For the administration of the questionnaire, each interview lasted between 15 and 30 minutes in total, and the physical examination took between 10 and 15 minutes [2].

Data were analyzed using SPSS software (version 26). The descriptive statistics, including frequency and percentage, mean, and standard deviation, were used to describe the sociodemographic and obstetrics characteristics, sexual function, anxiety, and depression. Some Data were compared among urban and rural using the $\chi2$ test, Fisher exact test, or Kruskal-Wallis test, as appropriate. Forward and backward regressions were employed based on P≤0.10. Statistical significance was considered to be P≤0.5. Data collection started only after approval of the research protocol by the Ethics Committee for Bimedical Research under the Faculty of Medicine and Pharmacy of Rabat. However, written consent to participate was obtained for all women. All study participants signed a informed consent form before the start of the study, interviews were conducted only after clarification and verbal consent, in a reserved room. Just women with age higher than 18 years participate in this study.

## Results

In the present study, 257 women were invited to participate. The mean age was 30 years, and mostly had a partner. We found also that 16.73 of the participants were experiencing their first pregnancy and more than 80% were multiparous. The majority of our study population had a Primary or less school level, and just 2.72% had a higher educational level (Table 1). Over one third of the population cannot read, and just 9.73% took more than 60 min to arrive from their house to the health service (Table 1). For self-reported health status of women, just 1.95%

**Table 1. Sociodemographic characteristics and clinical conditions of antenatal women (n = 257).**

| Characteristic | Variable | PNC (n = 257) | Characteristic | Variable | PNC (n = 257) |
|---|---|---|---|---|---|
| Maternal age | <20 | 13 (5) | Overall health rating | Very good | 5 (1.95) |
| | 20–34 | 155 (60) | | Good | 97 (37.74) |
| | >35 | 89 (35) | | Neither poor nor good | 115 (44.75) |
| Marital Status | Has partner | 253 (98.44) | | Poor | 36 (14.01) |
| | No partner | 4 (1.56) | | Very poor | 4 (1.56) |
| Education | Primary or less | 176 (68.48) | Have you been told you have anything wrong/ any conditions | No | 159 (61.87) |
| | Secondary | 74 (28.79) | | Yes | 98 (38.13) |
| | Higher | 7 (2.72) | Number of complications diagnosed | 0 | 53 (20.62) |
| Literacy | Cannot read | 89 (34.63) | | 1 | 103 (40.07) |
| | Can read parts of sentence | 105 (40.86) | | 2 | 89 (34.63) |
| | Can read whole sentence | 63 (24.51) | | 3–6 | 12 (4.66) |
| Employed | No | 233 (90.66) | Catégories of complications | Direct | 192 (74.70) |
| | Yes | 24 (9.34) | | Indirect | 65 (25.29) |
| Travel time to facility, minutes | <15 | 79 (30.74) | Types of direct complications | Gestational diabetes | 85 (33.07) |
| | 15–30 | 84 (32.68) | | Gestational hypertension | 23 (8.94) |
| | 30–60 | 69 (26.85) | | Urinary tract infection | 66 (25.68) |
| | >60 | 25 (9.73) | Types of indirect complications conditions | Vaginal infection | 32 (12.45) |
| Parity | 1 | 43 (16.73) | | Anemia | 44 (17.12) |
| | 2 to 4 | 197 (76) | | Syphilis* | 1 (0.38) |
| | ≥ 5 | 17 | | | |

reported very good health, more than 37.74% reported good health, and more than 30% of women had a health condition reported by the attending physician.

Based on the information provided, a list of conditions, classified them as direct and indirect, of which 20.23% had gestational diabetes, just 8.94% chronic hypertension, and urinary tract infection (25.68%) (Table 1).

Looking into use of substances, the majority of participants didn't use any type of substances during pregnancy. Using the WOICE tool to explore sex life, approximately half of the participants (49.03%) felt they were unsatisfied with their sex lives. Problems related to interest in sex (95.23%), pain during sex (42.06%), decreased vaginal lubrification (82.53%) and problems reaching orgasm (53.96%), are the reasons for sexual dissatisfaction (Table 2). The same table shows a significant relationship between urban and rural women ($p < 0.01$ for sexual satisfaction, $p < 0.001$ for decreased vaginal lubrification and $p = 0.042$ for problems reaching orgasm).

In order to investigate exposure to domestic and sexual violence, we asked participants "whether or not they were afraid of the current partner/ most recent spouse or any other person" if the spouse/ or any other person who pushed, hit and kicked". Just 13.22% of women reported to have suffered violence (Table 2).

As part of the Mental Health assessment of our study, we used the validated scales (PHQ-9 and GAD-7) [16, 17]. Abnormal conditions were considered if scores ≥10 [15, 18] and almost 48.64% of the women had anxiety symptoms, followed by 31% with depressive symptoms (Table 3). The distribution of anxiety was not significantly different by setting. However, there was a significant relationship between the distribution of depression between urban and rural

**Table 2. Social and sexual conditions among perinatal women.**

| Postpartum care | Urban (n = 156) | Rural(n = 101) | Total (n = 257) | p-value |
|---|---|---|---|---|
| **Substance use** | | | | |
| No | 155 (99.35) | 101 (100) | 256 (99.61) | **1** |
| Yes | 1 (0.64) | 0 (00) | 1 (0.39) | |
| **Exposure to violence** | | | | |
| No | 132 (84.62) | 68 (67.32) | 223 (86.77) | **0.82** |
| Yes | 24 (15.38) | 29 (28.71) | 34 (13.22) | |
| **Sexual Satisfaction** | | | | |
| No | 64 (41.03) | 62 (61.39) | 126 (49.03) | **<0.01** |
| Yes | 92 (58.97) | 39 (38.61) | 131 (50.97) | |
| **Reasons for sexual dissatisfaction (n = 126)** | | | | |
| Problem related to interest in sex | 78 (61.90) | 42 (33.33) | 120 (95.23) | **0.18** |
| Decreased vaginal lubrification | 64 (50.79) | 40 (31.74) | 104 (82.53) | **<0.001** |
| Problems reaching orgasm | 48 (38.09) | 20 (15.87) | 68 (53.96) | **0.042** |
| Pain during sex | 45 (35.71) | 8 (6.34) | 53 (42.06) | **0.017** |

women ($p<0.001$) (Table 3). The same table shows the significant relationship between sexual satisfaction and urban or rural areas ($p = 0.01$).

## Discussion

This study shows the results of the WHO-WOICE version 2.0 instrument used during antenatal care at 10 rural and urban health centers in Morocco. Overall, there was very good compliance, which shows that women are willing to participate in research during pregnancy, even when the research is about sensitive issues such as physical or sexual violence.

Based on the context provided, it seems that the WHO-WOICE had previously been used in a pilot study conducted in Jamaica, Kenya, and Malawi, which focused on low-risk populations in low-income settings [2]. However, the population in the current study conducted in Morocco was different in terms of sociodemographic data, with an older population who were more likely to have a partner compared to women in the pilot study countries [2]. Additionally, the current study reported a lower rate of substance use compared to the pilot study.

**Table 3. Mental health and sexual satisfaction among perinatal women in urban versus rural.**

| | ANC N = 257 | Urban (n = 156) | Rural (n = 101) | p-value |
|---|---|---|---|---|
| (a) Anxiety score Mean (SD) | 8.63 [± 6.11] | 9.10 [± 5.96] | 7.89 [± 6.31] | **0.82** |
| Score < 10 | 132 (51.36) | 81 (52) | 51 (50) | |
| Score ≥ 10 | 125 (48.64) | 75 (48) | 50 (50) | |
| (b) Depression score Mean (SD) | 7.29 [±5.49] | 7.04 [± 4.69] | 7.68 [± 6.55] | **<0.001** |
| Score < 10 | 175 (69) | 120 (77) | 55 (54) | |
| Score ≥ 10 | 82 (31) | 36 (23) | 46 (46) | |
| Sexual satisfaction | | | | **<0.1** |
| Oui | 131 (50.97) | 92 (59) | 39 (39) | |
| Non | 126 (49.03) | 64 (41) | 62 (61) | |

(a) GAD-7: seven items, with four-point scale: 0 (not at all) to 3 (Several days). A score ranging from 0 to 21 is considered a positive indicator for anxiety, equal to or greater than 10[17]. (b) PHQ-9: nine items, with a four-point scale: 0 (not at all) to 3 (several days), The score ranging from 0 to 27, considered a positive indicator of major depression, equal to or greater than 10[16]. It is estimated, as a positive indicator of major depression, equal to 10.

In this study, we observed a lower level of functioning rural relative to the urban area (65.34% in rural compared to only 14.74% in urban). Women with one clinically diagnosed health condition were generally more compared to those with no condition. When considering abnormal conditions evaluated by the WOICE instrument, it is indeed notable that more than 20% of women had no morbidities, and more than 40% had one morbidities. The current WHO maternal morbidity framework may not capture all the nonclinical and non-severe morbidities that women experience during pregnancy. This underreporting may occur because women may not report their symptoms or healthcare providers may not ask about them. Therefore, there is a need to give voice to women during care and actively ask about their symptoms to diagnose these morbidities. However, to fully understand the burden of maternal morbidity, healthcare providers should also use instruments during antenatal care that can bring to the surface underlying conditions. These instruments may include standardized questionnaires, physical examinations, and diagnostic tests that can detect morbidities that may not be obvious during routine care [11].

In perinatal visits, we can also evaluate **sexual health** of pregnant women, defined by the WHO as a "state of physical, mental, and social well-being in relation to sexuality". Sexual dysfunctions include little or no interest in sex, dyspareunia, and problems such as lubrication and genital sensation that can lead to sexual dissatisfaction and subsequent sexual inactivity [12, 13]. There is a well-documented decrease in sexual activity during pregnancy. There are several reasons for this, including maternal morbidities such as vaginal discomfort, nausea, and fatigue, as well as physiological changes such as hormonal fluctuations. In addition, cultural beliefs and lack of information about sexuality during pregnancy can also contribute to a decrease in sexual activity. Some people may believe that sex during pregnancy is harmful to the baby, while others may view it as unappealing or taboo [12]. This study reported that **sexual satisfaction** was less during pregnancy [12]. Our findings showing also that most participants (49.03%) reported being unsatisfied with their sex lives. Another study found a significant decrease in sexual satisfaction in rural versus urban areas (p = 0.002) [14]. This result corroborates those of our study which showed that sexual satisfaction also was significantly among urban women than rural women (41% versus 61%, respectively; P<0.01).

Nevertheless, only 2.33% of these women stopped having sex as soon as they became pregnant, which means that some of the women surveyed did not need sex but do it to satisfy their husbands. Among pregnant women, sexual satisfaction is related to the woman's acceptance of her body image, the type of communication with her partner, and having sex [12].

**Violence** against women is a pervasive problem that affects women all over the world, regardless of culture or socioeconomic status. One of the reasons it can be difficult to approach and identify is because many acts of violence are considered normal or acceptable in certain cultures, making it harder to recognize them as abusive or harmful [15]. Fear is also a major factor that can prevent women from reporting incidents of violence, and can also make health professionals reluctant to intervene. Victims of violence may fear retribution from their abuser, or may worry that they won't be believed or taken seriously. Health professionals may fear legal or professional consequences if they intervene in cases of suspected abuse, or may be uncertain about how to respond to these situations [18–21]. Our study showed that 12.23% of women had been exposed to some kind of violence, with a difference between urban and rural areas, compared to 12.8% in the pilot study [2] and 8% and 11% in previous reports in urban and rural regions in Brazil, as evidenced by a multicenter study on violence against women published by the WHO in 2005 [18]. Other studies have shown a prevalence of domestic violence greater than 40% [22, 23]. It is likely that our study obtained under estimated data since women did not consider themselves to be victims of violence, despite the ill treatment [24, 25].

Looking for mental health and sexual satisfaction among perinatal women in urban versus rural, a statistically significant differences between rural and urban women in depression (23% versus 46%, respectively; p<0.001). In contrast, there were no statistically significant difference between rural and urban women who had experienced anxiety (48% versus 50%, respectively; P = 0.82). In this sense, a studies [26, 27] has shown that Risk for depression and/or anxiety was found to be higher in the rural group.

One of the strengths of the present study is its large sample size and random sampling that increased the generalizability of the study results. Also, this study's cross-sectional nature is one of the limitations of the design that the relationships shown do not exactly indicate a causal relationship.

## Conclusion

The maternal health agenda is undergoing a paradigm shift from preventing maternal deaths to promoting women's health and wellness. The findings of the pilot study also highlight that the focus of routine ANC could be broadened beyond the care of chronic conditions to also include a simple way of screening for intimate partner violence and mental health conditions. WHO has long recognized that integration of antenatal care with other health services is a key strategy to reduce missed opportunities for patient contact, and to effectively address the comprehensive health needs of women. Anxiety, depression, impaired functioning, and violence are frequent conditions among pregnant women. These issues are brought to light by the use of the WHO-WOICE instrument and merit further prioritization to improve women's health.

## Acknowledgments

We would like to express our deep gratitude to all the participants in this study, parturient, midwives and nurses, managers and officials at the level of the Ministry of Health.

## Author Contributions

**Conceptualization:** Hanane Hababa.

**Formal analysis:** Hanane Hababa.

**Investigation:** Hanane Hababa.

**Methodology:** Hanane Hababa.

**Project administration:** Hanane Hababa.

**Resources:** Hanane Hababa.

**Software:** Hanane Hababa.

**Validation:** Hanane Hababa, Bouchra Assarag.

**Visualization:** Hanane Hababa, Bouchra Assarag.

**Writing – original draft:** Hanane Hababa.

**Writing – review & editing:** Hanane Hababa.

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
