## [Decision Letter · Decision Letter 0]

5 Feb 2023

PONE-D-22-22581

Overview of maternal morbidity in Morocco Marrakech-Safi region

PLOS ONE

Dear Dr. HABABA,

Thank you for submitting your manuscript to PLOS ONE. After careful consideration, we feel that it has merit but does not fully meet PLOS ONE’s publication criteria as it currently stands. Therefore, we invite you to submit a revised version of the manuscript that addresses the points raised during the review process.

The reviewer feels that the methodology presented in the manuscript is not described in sufficient detail. They note that throughout the manuscript references are missing and some of the abbreviations used are not explained. In addition, the reviewer suggests that the manuscript undergoes copyediting/proofreading.

Please note that we have only been able to secure a single reviewer to assess your manuscript. We are issuing a decision on your manuscript at this point to prevent further delays in the evaluation of your manuscript. Please be aware that the editor who handles your revised manuscript might find it necessary to invite additional reviewers to assess this work once the revised manuscript is submitted. However, we will aim to proceed on the basis of this single review if possible. 

We look forward to receiving your revised manuscript.

Kind regards,

Alex Schaefer, PhD

Associate Editor

PLOS ONE

and https://journals.plos.org/plosone/s/file?id=ba62/PLOSOne_formatting_sample_title_authors_affiliations.pdf.

A clean copy of the edited manuscript (uploaded as the new *manuscript* file).

5. Please amend either the title on the online submission form (via Edit Submission) or the title in the manuscript so that they are identical.

7. Please include a caption for figure 1.

Reviewers' comments:

Reviewer's Responses to Questions

**Comments to the Author**

1. Is the manuscript technically sound, and do the data support the conclusions?

Reviewer #1: No

2. Has the statistical analysis been performed appropriately and rigorously? 

Reviewer #1: No

3. Have the authors made all data underlying the findings in their manuscript fully available?

Reviewer #1: Yes

4. Is the manuscript presented in an intelligible fashion and written in standard English?

Reviewer #1: No

5. Review Comments to the Author

Reviewer #1: General comments:

This is an interesting study that aims to describe the less severe maternal morbidity in Morocco in pregnant and recently delivered women using tools developed by the WHO. This is a unique study in that there are few studies focusing on less severe maternal morbidity, particularly in Morocco. However, the methods are not described in sufficient detail. There are some missing references throughout the manuscript. Some abbreviations are defined in the text while others are not. The manuscript can also benefit from some language editing to make things clearer.

Introduction:

1. The introduction provides a very detailed background on maternal mortality and can be made more concise and streamlined according to the objective.

2. The last paragraph of the introduction states, “This study will test the feasibility and acceptability of using a tool to measure the impact of pregnancy and childbirth on women's health status. Once validated in our context…”. However, the objective of this study is, “to describe the current state of maternal morbidity in the prefecture of Marrakech.” This study is neither testing the feasibility and acceptability of the tool, nor is it validating it in Morocco.

Methods:

1. More detailed description of the study design and sites should be included. For example, which ten health centers were included? How were these centers selected? How many were in rural and how many were in urban areas?

2. What was the sampling technique used to recruit participants in the study?

3. Please mention details about the informed consent process.

4. It is mentioned in the first paragraph that, “the time allocated to the questionnaire (45 to 65 minutes)…”. The second paragraph says that, “each interview with the woman lasted between 15 and 30 minutes in total.” What was the reason that interviews only took about half the suggested time?

5. “The data analysis was done with the two software programs Epi Info 7 for the quantitative data and Invivo 10 was used for the qualitative data.” What kind of qualitative data were included? How were the quantitative and qualitative data analyzed?

Results:

1. When comparing the proportion of a characteristic between two groups (for example, anxiety and depression in women during PNC and PoNC), the statistical significance should be reported.

2. Tables 1 and 2 should be combined.

3. The tables are not referenced in the text which is a little difficult to follow.

4. “Nevertheless, 39.29% in PNC and 37.94% in PoNC declared….”. Table IV says that 44.75% women in the PNC consultation declared that their health status was neither good nor bad.

5. “Using both tools, it was revealed that PNC women were more likely to suffer from anxiety (83.65% vs. 29.24% in PNC) and also signs of depression (43% vs. 17.78% in PNC).” It is unclear what the percentages in the parentheses represent. Additionally, what were the signs of depression? Table V: Mental health only describes symptoms of anxiety and not depression.

6. In table V: Mental health, what does the no and yes below item no. 7 on the anxiety score refer to?

7. In table VII, the numbers in the row “number of complications diagnosed” are not equal to the number of women who received prenatal consultation for both urban and rural women. There is a similar issue with the “number of complications diagnosed” for urban women in the postnatal consultation. There appears to be some missing data which should be mentioned.

8. In table VII, the numbers in the subcategories of types of indirect complications in the PNC rural column equal to 55, when there were 59 women who experienced indirect complications. Similarly, the direct complications in the same column equal to 33, when there were 58 women who experienced direct complications.

9. In table VII, there are 32 women with back pain in the PNC urban column, when there were 28 women in this group who experienced indirect complications.

10. Some of the headings in table VII are in French.

11. The results do not show the findings from the qualitative part of the study.

Discussion:

1. The recommendations are considerably longer than the discussion. The recommendation should be shortened and included within the discussion section, keeping in view the PLOS ONE format for a research article.

2. “In addition, our study found that the prevalence of emotional abuse was more frequent than physical abuse.” This finding is not shown in the results section.

3. “This result does not corroborate with a study conducted in Egypt in 2017 [18], which showed that women in rural areas are more exposed to domestic violence than those in urban areas.” The results state that, “…with rates varying by setting from 12% in urban areas to 29% in rural areas.” This indicates that the findings of the present study do corroborate with the study conducted in Egypt.

4. Some of the recommendations are beyond the scope of this study. For example, the impact of seeing pregnancy as an opportunity, adopting a rights-based approach, and premarital consultation were not evaluated in the study. Therefore, these recommendations cannot be directly derived from the results of this study.

5. It would be good to comment on the generalizability of the findings of this study.

Conclusion:

1. Overall, the conclusion seems generic and not specific to the salient findings of the study and their broader implications.

2. Typically, there are no references in the conclusion.

3. “Although women interviewed report a high level of satisfaction with MM tools…”. This statement is not supported by this study.

6. PLOS authors have the option to publish the peer review history of their article (what does this mean?). If published, this will include your full peer review and any attached files.

Reviewer #1: **Yes: **Dr Shabina Ariff

---

## [Author Response · Author response to Decision Letter 0]

21 Mar 2023

All the relevant remarks you sent are corrected, one by one. In order to ensure that the data presented support the conclusions, and to present the article appropriately, the article has been revised and the main results have been presented.

---

## [Editor Report · Decision Letter 1]

29 May 2023

PONE-D-22-22581R1Overall Maternal Morbidity during Pregnancy using the WHO-WOICE toolsPLOS ONE

Dear Dr. HABABA,

Thank you for submitting your manuscript to PLOS ONE. After careful consideration, we feel that it has merit but does not fully meet PLOS ONE’s publication criteria as it currently stands. Therefore, we invite you to submit a revised version of the manuscript that addresses the points raised during the review process.

We look forward to receiving your revised manuscript.

Kind regards,

José Paulo de Siqueira Guida, PhD

Academic Editor

PLOS ONE

Journal Requirements:

Additional Editor Comments:

Dear author,

thanks for the submission of this new version of the article.

I still have a few questions:

1) authors described in their methods that they have applied the WOICE tool, regarding functioning, however I could not find the description of the results. Consider including it.

2) In table 3, authors included sexual satisfaction in French, correct it.

3) Why did authors choose to compare rural and urban settings for anxiety, depression and sexual satisfaction, but not for the other variables included in tables 1 and 2? Presentation of results must be uniform throughout the article. Discuss the best way to present results. I believe that the comparison between both settings is interesting, and if authors decide to do it, they should uniformize the analysis in the other tables.

---

## [Author Response · Author response to Decision Letter 1]

24 Jul 2023

1-The WOICE tool is made up of several sections. This article focuses on results from the prenatal period.

2-Already corrected

3-The rural/urban comparison has been added to table 2 and the discussion. Table 1 presents the general characteristics, and the authors do not consider it relevant to add data from both rural and urban areas.

---

## [Editor Report · Decision Letter 2]

3 Aug 2023

Overall Maternal Morbidity during Pregnancy using the WHO-WOICE tools

PONE-D-22-22581R2

Dear Dr. HABABA,

We’re pleased to inform you that your manuscript has been judged scientifically suitable for publication and will be formally accepted for publication once it meets all outstanding technical requirements.

Kind regards,

Aklilu Habte Hailegebireal, MPH

Academic Editor

PLOS ONE

Additional Editor Comments (optional):

Reviewers' comments:

<quillbot-extension-portal></quillbot-extension-portal>

---

## [Editor Report · Acceptance letter]

7 Aug 2023

PONE-D-22-22581R2 

Overall Maternal Morbidity during  Pregnancy using the WHO-WOICE tools 

Dear Dr. HABABA:

I'm pleased to inform you that your manuscript has been deemed suitable for publication in PLOS ONE. Congratulations! Your manuscript is now with our production department. 

Kind regards, 

on behalf of

Dr. Aklilu Habte Hailegebireal 

Academic Editor

PLOS ONE